# Cost-Utility of Attachment-Based Compassion Therapy (ABCT) for Fibromyalgia Compared to Relaxation: A Pilot Randomized Controlled Trial

**DOI:** 10.3390/jcm9030726

**Published:** 2020-03-07

**Authors:** Francesco D’Amico, Albert Feliu-Soler, Jesús Montero-Marín, María T. Peñarrubía-María, Mayte Navarro-Gil, William Van Gordon, Javier García-Campayo, Juan V. Luciano

**Affiliations:** 1Personal Social Services Research Unit, London School of Economics and Political Science, London WC2A 2AE, UK; damico@lse.ac.uk; 2Institut de Recerca Sant Joan de Déu, 08950 Esplugues de Llobregat, Spain; a.feliu@pssjd.org; 3Teaching, Research & Innovation Unit, Parc Sanitari Sant Joan de Déu, 08830 St. Boi de Llobregat, Spain; 4Faculty of Psychology, Autonomous University of Barcelona, 08193 Cerdanyola del Vallès, Spain; 5Department of Psychiatry, University of Oxford, Warneford Hospital, Oxford OX3 7JX, UK; jesus.monteromarin@psych.ox.ac.uk; 6PHC Bartomeu Fabrés Anglada, DAP Baix Llobregat Litoral, Unitat Docent Costa de Ponent, Institut Català de la Salut, 08850 Gavà, Spain; 32829mpm@comb.cat; 7Centre for Biomedical Research in Epidemiology and Public Health, CIBERESP, 28029 Madrid, Spain; 8Department of Psychology and Sociology, University of Zaragoza, 50009 Zaragoza, Spain; maytenavarrogil@gmail.com; 9Centre for Psychological Research, University of Derby, Derby DE22 1GB, UK; w.vangordon@derby.ac.uk; 10Miguel Servet Hospital, Aragon Institute of Health Sciences (I+CS), 50009 Zaragoza, Spain; jgarcamp@gmail.com; 11Primary Care Prevention and Health Promotion Research Network, RedIAPP, 28029 Madrid, Spain

**Keywords:** attachment-based compassion therapy, mindfulness, cost-utility, economic evaluation

## Abstract

A recent study has supported the efficacy of Attachment-Based Compassion Therapy (ABCT) compared to relaxation (REL) for the management of fibromyalgia (FM). The main objective of this paper is to examine the cost-utility of ABCT compared to REL in terms of effects on quality-adjusted life years (QALYs) as well as healthcare costs. Forty-two Spanish patients with FM received 8 weekly group sessions of ABCT or REL. Data collection took place at pre- and 3-month follow-up. Cost-utility of the two treatment groups (ABCT vs. REL) was compared by examining treatment outcomes in terms of QALYs (obtained with the EQ-5D-3L) and healthcare costs (data about service use obtained with the Client Service Receipt Inventory). Data analyses were computed from a completers, ITT, and per protocol approach. Data analysis from the healthcare perspective revealed that those patients receiving ABCT exhibited larger improvements in quality of life than those doing relaxation, while being less costly 3 months after their 8-week treatment program had ended (completers: incremental cost M, 95% CI = €−194.1 (−450.3 to 356.1); incremental effect M, 95% CI = 0.023 QALYs (0.010 to 0.141)). Results were similar using an ITT approach (incremental cost M, 95% CI = €−256.3 (−447.4 to −65.3); incremental effect M, 95% CI = 0.021 QALYs (0.009 to 0.033)). A similar pattern of results were obtained from the per protocol approach. This RCT has contributed to the evidence base of compassion-based interventions and provided useful information about the cost-utility of ABCT for FM patients when compared to relaxation. However, the small sample size and short follow-up period limited the generalizability of the findings.

## 1. Introduction

### 1.1. Loving-Kindness Meditation and Compassion-Based Interventions

Under the umbrella term “fourth wave” [1], there is a group of therapies with their origin in European existential psychotherapy and related humanistic approaches that go beyond symptom improvement to enhancing well-being. These approaches include loving-kindness and compassion meditation, as well as related approaches such as positive psychology programs, dignity- and gratitude-promoting approaches, meaning-centered therapy, forgiveness-oriented techniques, and spiritually-informed therapies [1]. Loving-kindness meditation (LKM; metta), compassion meditation (CM; karuna), sympathetic joy (muditā), and equanimity (upekkha) are the four immeasurables that can be cultivated during meditation practice. Practices of both LKM and CM are inherent to all Buddhist traditions, but have particular prominence in Mahayana Buddhist schools [2]. The main aim of LKM is to develop benevolence and warmth toward all sentient beings, whereas CM includes techniques to cultivate a deep, genuine sympathy for those that are suffering, together with a commitment to alleviate and prevent suffering [3,4,5,6]. LKM and compassion-based interventions (CBIs) have been shown to be effective for treating a considerable range of mental health issues in both clinical and healthy adult and non-adult populations [2].

There are seven empirically supported CBIs that have been the subject of evaluation in RCTs [7]: Cognitively-Based Compassion Training [8], Cultivating Emotional Balance [9], Mindful Self-Compassion [10], the Compassion Cultivation Training [11], Compassion-Focused Therapy [12], Compassion and Loving-Kindness Meditations [5], and Attachment-Based Compassion Therapy [13]. A meta-analysis of some of the CBIs listed above [14] showed that compared to wait-list control conditions, CBIs elicited a significant moderate effect size for improving compassion (d = 0.55), self-compassion (d = 0.70), mindfulness (d = 0.54), depression (d = 0.64), anxiety (d = 0.49), psychological distress (d = 0.47), and well-being (d = 0.51). The same meta-analysis showed that compared to active control conditions, there was a significant moderate effect size for improvements in self-compassion (d = 0.60), mindfulness (d = 0.46), depression (d = 0.62), anxiety (d = 0.42), psychological distress (d = 0.40), and well-being (d = 0.48). Furthermore, Montero-Marin and colleagues [15] showed that ABCT as a coadjuvant of usual care produced greater improvements in the functional status of patients with fibromyalgia (FM) compared to relaxation practices at post-treatment (d = 1.33) and 3-month follow-up (d = 1.38). Significant improvements in the ABCT group were also observed in the following secondary outcomes at post-treatment and follow-up: clinical severity (d = 0.96 and d = 1.14), anxiety (d = 1.03 and d = 0.90), depression (d = 0.94 and d = 1.01), quality of life (d = 0.84 and d = 0.85), and psychological flexibility (d = 1.13 and d = 1.18).

### 1.2. The Importance of Economic Evaluations in Healthcare

As a consequence of population growth as well as a larger elderly population living with multimorbidity and disability, healthcare and societal costs have steadily risen over recent decades in industrialized countries [16]. These rising costs threaten the sustainability of healthcare systems, and force policy-makers and financial stakeholders to make difficult decisions on how to allocate scarce resources [17]. Most public healthcare systems use economic evaluations as decision-making criteria [1]. Economic evaluations (cost-effectiveness, cost-utility, cost-benefit, cost-minimization, and cost-consequences analysis), in which costs and effects of two or more healthcare interventions are compared, can support policy-makers in such allocation decisions [18]. Cost-effectiveness and cost-utility analysis are the most common analytic techniques employed by health economists [19,20,21]. In cost-effectiveness analysis, consequences are valued in terms of a single measure of health outcome (e.g., points in an anxiety scale); whereas in cost-utility analysis, the outcome is the cost per quality-adjusted life-year (QALY; combination of quantity and quality of life).

### 1.3. Why It Is Important to Know the Cost-Utility of ABCT

Although most CBIs were designed ten or more years ago, to our knowledge, no economic evaluation of these therapies using RCTs has been conducted to date. However, the burgeoning interest in these relatively new therapies among clinicians and researchers, as evidenced by aforementioned meta-analyses of clinical outcomes [3,14], suggests a need for demonstrating whether CBIs should be acknowledged as an option by policy-makers or stakeholders when deciding which treatment is the most efficient for a particular patient group. The assessment of economic impact of CBIs is a research priority [1,22]. As in the case with acceptance and mindfulness-based interventions [23,24], to evaluate the efficiency of CBIs as standalone or concomitant treatments is an issue of crucial importance for the implementation of these therapies in real-world clinical practice.

ABCT seems well-suited for treating FM syndrome, characterized by widespread chronic pain and a constellation of concomitant physical and mental symptoms, including excessive self-criticism [25], a lower secure attachment style and higher avoidant and anxious-ambivalent attachment styles in comparison with healthy adults [26]. However, the efficiency of ABCT as a FM treatment has not been previously analyzed. In this study, we extend the results of Montero-Marin et al. [15] by examining the 3-month cost-utility of ABCT compared to Relaxation in terms of gains in QALYs from a healthcare perspective. ABCT and Relaxation were structurally equivalent, which provides a comparison of ABCT to a suitable active control that matches it in non-specific factors, but does not contain compassion- and mindfulness-based components.

## 2. Materials and Methods

### 2.1. Design

A detailed description of the RCT protocol is provided elsewhere [15]. A RCT with 2 treatment arms and a pre-, post- and 3-month follow-up assessment was conducted. Adults suffering from FM were allocated to treatment-as-usual (TAU) + ABCT or TAU + Relaxation training using a parallel assignment and a computer-generated randomization list. The economic evaluation alongside this pilot RCT was conducted according to the CHEERS statement [27] and the Good Research Practices for Cost-Effectiveness Analysis Alongside Clinical Trials [28]. The study followed the Helsinki Convention norms and subsequent updates. The study protocol was approved by the ethical review board of the regional health authority of Aragon, Spain (PI15/0049; 01/04/2015). Findings from a third treatment arm outlined in the protocol (NCT02454244) that received mindfulness plus amygdala retraining are to be reported elsewhere. Specifically, the cost-utility analysis reported here was not registered in ClinicalTrials.gov, therefore this should be considered an exploratory analysis. For transparency and analytical reproducibility purposes, SPSS data and STATA syntax can be accessed at OSF: https://osf.io/zfjcr/.

### 2.2. Participants

Forty-two patients with FM were recruited in general practices from Aragon, Spain. Recommendations for pilot studies propose that the dataset includes approximately 15–20 participants in each arm in order to estimate parameters when the expected standardized effect size is medium [29]. The inclusion criteria were: (1) male or female aged between 18 and 65 years; (2) able to read and understand Spanish; (3) diagnosed with FM by a rheumatologist working for the Spanish National Health Service following the ACR 1990 criteria [30]. The exclusion criteria were: (1) presence of a severe Axis I psychiatric disorder (dementia, schizophrenia, paranoid disorder, alcohol and/or drug use disorder) or severe somatic disorder that from the clinician’s point of view, prevented the patient from effectively completing a psychological assessment; or (2) concurrent participation in another RCT. Medication use was not an exclusion criterion, as long as the participant agreed not to modify the pharmacological treatment along the study period.

### 2.3. Procedure

General practitioners recruited potential participants from January to March 2015. Once the required sample size was achieved, the participants were interviewed at the primary care center by a clinical psychologist, who confirmed suitability according to the aforementioned inclusion criteria and provided a general overview of the study. Participants provided written consent and were informed that they could withdraw from the study at any time without it affecting the quality of the treatment delivered by their GP. The randomization occurred in April 2015 and the ABCT and Relaxation treatments were delivered from May to October 2015. An independent research assistant generated the random allocation sequence to determine group assignment. Allocation details were concealed from the research group until all participants had been randomly assigned. Participants agreed to participate prior to randomization and were informed of group allocation after baseline assessment. A clinical psychologist, not involved in the study and blind to participant allocation, was the outcome assessor.

### 2.4. Treatments

Both ABCT and Relaxation were coadjuvants of treatment as usual (TAU) provided by healthcare professionals of the public National Health System for FM patients. TAU is offered by the corresponding GP or rheumatologist and usually consists of pharmacological treatment. General practitioners may refer the FM patient to any specialist when necessary.

#### 2.4.1. ABCT

It is a compassion-based program that was slightly adapted for FM patients [13,31]. It focused on augmenting the patient’s ability to be considerate and kind towards (i) themselves and their own suffering experience, and (ii) others’ experience of suffering. ABCT comprises 8 weekly 2-hour sessions plus 3 booster monthly sessions. It includes formal practices of mindfulness and visualizations based on self-compassion and the attachment style that was generated in childhood. The program includes daily homework assignments that take approximately 15–20 min to complete. The ABCT instructor was a psychologist with accredited experience who delivered the intervention using a group format (up to 12 participants per group).

#### 2.4.2. REL

Relaxation training was the active control condition. Participants allocated to this treatment received a low-intensity and non-specific intervention lasting 8 weekly 2-hour sessions that included some relaxation techniques such as imagery, progressive muscle relaxation, autogenic training, etc. The program includes daily homework assignments that take approximately 15–20 min to complete. After completing the 8 weekly sessions, participants received 3 booster sessions. Relaxation training has been shown to improve FM symptoms [32]. For ethical reasons, following completion of the study, participants in this study arm had the option to receive ABCT. A psychologist with experience on relaxation training delivered this intervention using a group format (up to 12 participants per group).

A session-by-session description of ABCT and Relaxation training is provided elsewhere [15].

### 2.5. Study Measures

#### 2.5.1. Sociodemographic-Clinical Questionnaire

It collected the following information: gender, age, marital status, living arrangement, education level, work status, family and personal medical history, years of FM diagnosis, and comorbid illnesses.

#### 2.5.2. The EuroQoL Questionnaire (EQ-5D-3L)

It is a generic measure of health-related quality of life that comprises two parts [33]: A 5-domain descriptive system that evaluates level of mobility, self-care, usual activities, pain-discomfort, and anxiety-depression. Each domain is described at three levels: ‘no problems’ (level 1), ‘some problems’ (level 2), and ‘extreme problems’ (level 3). The time frame is the day of responding. Combinations of these categories define a total of 243 unique health states. Part 2 captures the present subject’s health on a Visual Analogue Scale (0 to 100), where the respondent can self-report their current health status, 100 being the best possible health level.

#### 2.5.3. The Client Service Receipt Inventory (CSRI)

The CSRI version used here was developed to collect retrospective data related to medications and service use [34]. Regarding medication use, we requested information about specific prescribed medications related to FM (analgesics, opioids, anticonvulsants, antidepressants, etc.), including the name of the drug, the prescriber, the dosage level, the total number of prescription days, the daily dosage consumed, the reasons for changing the drug (when applicable), and adherence. Concerning service use, we collected information about emergency services (total visits), general medical inpatient hospital admissions (total days), and outpatient health care services (total visits to GP, nurse, social worker, psychologist, etc.). We recorded whether each service was being provided by the public or by the private healthcare sector. We also wanted to know the type and number of diagnostic tests administered. The CSRI was administered on two occasions (at baseline and at 3-month follow-up) with different timeframes; the previous year at baseline and the previous 3 months at follow-up.

### 2.6. Data Analyses

#### 2.6.1. Description of The Costing Procedure from The Healthcare Perspective

We calculated direct health care costs by summing the costs derived from medication, medical tests, use of health-related services, and cost of the staff delivering the ABCT and Relaxation training. In the case of medication, we calculated the price per milligram (mg) following the Vademecum International (Red Book; edition 2016) and included value-added tax. The total costs of FM-related medications were calculated by multiplying the price per mg by the daily dosage used (in mg) and the number of days that the treatment was received. The main source of the unit cost data for medical tests and health services use was the SOIKOS database (http://esalud.oblikue.com/). The calculation of the total costs of the ABCT and REL treatments was based on the price per participant per group session of a clinical psychologist, indicated by the Spanish Official College of Psychology. The cost of ABCT and Relaxation was assumed to be consistent across all sessions and groups, but the number of patients attending those sessions was not. Therefore, treatment costs were dependent on the number of sessions attended by each patient. Unit costs are expressed in Euros (€) based on 2016 prices. Table 1 shows the unit costs of healthcare resources. It was not necessary to apply a discount factor to the costs because the time horizon was less than a year.

#### 2.6.2. Utility Scores

They were obtained from the EQ-5D-3L and were computed to rate patients’ HRQoL from 0 (as bad as death) to 1 (perfect health). They reflect how the general population values the health status described by the individual, which is preferred for economic evaluations from a broad perspective. QALYs were obtained on the basis of these scores using the Spanish tariffs of EQ-5D-3L [35]. QALYs are a measure which takes into account both disease-burden and mortality, and can be used to provide a common metric to assess the extent of the benefits gained from different treatments in terms of HRQoL and survival for the patient. A QALY places a weight on time in different health states. A year of perfect health is worth 1 and a year of less than perfect health is worth less than 1.

#### 2.6.3. Cost-Utility Analyses

We computed incremental cost-utility ratios (ICUR), defined as the ratio between incremental costs and incremental effects (QALYs). The ICUR can be interpreted as additional costs associated with realizing one additional QALY compared to the patients receiving relaxation. There are four potential scenarios:(i)ABCT costs less and is more effective than Relaxation;(ii)ABCT costs more and is less effective than Relaxation;(iii)ABCT costs less but is less effective than Relaxation;(iv)ABCT costs more and is more effective than Relaxation.

Scenarios (i) and (ii) exhibit strong dominance, and the decision of whether or not to adopt the target intervention is typically straightforward. However, in the scenarios (iii) and (iv), the decision will depend on the value attached to differences in outcome. In the cost-utility analysis, incremental costs and incremental effects were estimated computing Seemingly Unrelated Regressions. Using this methodology, cost and outcome measures were included in a bivariate system that implemented a regression of costs and QALYs on treatment allocations, i.e., whether they were assigned to ABCT or Relaxation. The regressions controlled for the following baseline variables: treatment group, gender, age, marital status, living arrangements, education level, work status and number of treatment sessions attended. Estimates were run using 1000 bootstrap replications to address a possible skewness in the distribution of the dependent variables [36]. Firstly, we computed a complete case analysis. Seven FM patients without 3-month follow-up data were excluded. Secondly, the analysis was recomputed following an ITT approach (first sensitivity analysis). In this case, missing values (16.7%) were imputed. Specifically, we used multiple imputation methods (chained equations approach) to impute missing values for the EQ-5D-3L domains at follow-up. The imputation model, run on 5 imputed datasets, used the available information on socio-demographic characteristics and service use associated with the missing information. Finally, a per protocol analysis (second sensitivity analysis) was computed on a sample that included only those patients who attended ≥6 treatment sessions out of 8, with a final sample size of 34 patients. In the ITT case, given that the small size of the sample made the bootstrap model difficult to converge, we estimated the cost-utility model adopting a classical SUR approach with a small-sample adjustment for the standard errors. Statistical analyses were carried out using SPSS v22.0 (SPSS Inc., Chicago, IL, USA) and STATA v15.0 (StataCorp, College Station, TX, USA).

## 3. Results

### 3.1. Participant Characteristics, Flow and Compliance

Overall, 83 participants were referred to the RCT: 64 met the inclusion criteria, resulting in 23 receiving ABCT and 19 REL. As stated before, findings from a third treatment arm (mindfulness + amygdala retraining) are to be reported elsewhere. The attrition rate was low because 20 participants (87%) from the ABCT arm and 15 (79%) from the REL arm completed the 3-month follow-up assessment, respectively. No selective dropout was observed. The initial socio-demographic characteristics of the patients in the two treatment arms were fairly similar (see Table 2). At baseline, we did not find statistically significant differences in age, marital status, dwelling, educational level, or work status. Most patients assigned to ABCT or REL complied with optimal attendance rates. Specifically, 83% of ABCT participants and 95% of REL participants attended six or more treatment sessions. Figure 1 shows the flow of participants through the economic evaluation.

### 3.2. Direct Costs and QALYs

Table 3 displays the estimated total mean direct health-related costs per patient over a period of one year (baseline) and over a period of three months (follow-up). The last columns of the table presents the *p*-values of the Wald test, under the hypothesis of equality between the intervention groups and effect sizes (Cohen’s *d*). The test is adjusted for the main socio-demographic variables and for the number of treatment sessions attended. At baseline, the adjusted *p*-value shows that there were no significant differences in direct costs between the two treatment groups. The intervention associated costs in the ABCT group were significantly higher than in the REL group (small effect size) due to different attendance rates of the two interventions, considering that the individual costs of the sessions were of a similar size. No other significant differences were found between the two treatment groups with regard to the mean total direct costs per patient at follow-up (i.e., including the costs due to primary care visits, specialized healthcare visits, etc.), nor was there a significant difference in QALYs between the treatment groups. The EQ-5D utility scores for the ABCT group showed improvement during the study period but did not differ significantly between the groups.

### 3.3. Cost-Utility of ABCT Compared to Relaxation

As shown in Table 4, ABCT obtained “dominant” ICERs for direct medical costs per QALY gained from three statistical approaches (completers, ITT, and per protocol). The comparison of the direct medical costs, ABCT vs. REL, shows that the former is superior to the latter (with non-significant lower costs but significant better outcomes), with an incremental cost of €−194.1 and an incremental outcome of 0.023 QALY, yielding an ICUR value of €−8,439 per QALY gained from the completers approach. In the first sensitivity analysis (ITT approach), we recalculated the costs and QALYs by computing imputation of missing values. Again, the same superiority applies (ABCT significantly achieved more QALYs at lower costs), with ABCT achieving an incremental cost of €−256.3 and an incremental outcome of 0.021 QALY, yielding an ICUR value of €−12,204 per QALY gained. Results in the second sensitivity analysis (per protocol) were in the same direction, but the incremental cost and effect were not statistically significant. In sum, ABCT was more cost-effective than relaxation, with the outcome being significantly better for ABCT than for REL both in the completers and in the ITT approaches, whilst the costs were significantly lower for ABCT only within the ITT approach.

## 4. Discussion

### 4.1. Summary of Key Findings

To the authors’ knowledge, this study reports the first cost-utility analysis of a CBI in the Spanish public healthcare system. Findings demonstrate that ABCT was more cost-effective for the Spanish healthcare system than relaxation. In Spain, a threshold of €22,000–25,000 per QALY gained is found to be consistent with decisions of adopting new technologies by the National Health Service [37], therefore, ABCT might be a potential cost-effective option for FM within the context of public healthcare practices. This is despite the fact that we did not find significant differences in costs between the treatment groups or in utility scores/QALYs between the groups at follow-up.

### 4.2. Comparison with Relevant Findings from Previous Published Studies

The promising results obtained in the present study in terms of effects in quality of life and costs are in line with some previously reported economic evaluations of mindfulness- and acceptance-based interventions in studies carried out by independent research groups in different countries [23,24]. For instance, Kemani and colleagues [38] compared the cost-effectiveness of Acceptance and Commitment Therapy versus applied relaxation (AR) for Swedish patients with unspecific, longstanding pain, both treatments being delivered for three months. The economic evaluation revealed that ACT was more cost-effective than AR at post-treatment and at three-month follow-up assessment. More recently, Luciano et al. [39] reported that a group-based form of ACT was cost-effective for FM patients recruited in Spanish general practices in comparison to recommended pharmacological treatment (pregabalin plus duloxetine in case of concomitant major depression) regardless of the economic perspective (healthcare or societal).

In the USA, a study compared the cost-utility of MBSR, CBT, and treatment-as-usual for the management of chronic low back pain from both the payer and societal perspectives [40]. Costs from the payer perspective included overall healthcare costs. Costs from the societal perspective included participant copayments for healthcare, employer productivity losses, and overall healthcare costs to the health plan. Data analyses from the societal perspective yielded an ICER of $−21,294 per QALY for the comparison MBSR vs. TAU. Taking the USA willingness-to-pay threshold into account, MBSR had a 90% probability of being less than $50,000/QALY, whereas CBT had an 81% probability of being less than $50,000/QALY. Additionally, of interest, is a study that compared MBCT to a waiting-list in Danish women treated for breast cancer that suffered from persistent pain [41]. Their analyses were conducted from the health care system perspective and included data on health care utilization and pain medication retrieved from national registries for the period from baseline to six months post-treatment. As expected, the authors indicated lower costs and greater pain reduction in those patients receiving MBCT compared to waiting-list. Recently, Navarro-Gil and colleagues [31] carried out a non-randomized RCT comparing ABCT with a waiting list (WL) control group. In this study conducted with healthy subjects, ABCT was significantly more effective than WL for increasing self-compassion, mindfulness, and a secure attachment style. Moreover, ABCT managed to reduce psychological distress and experiential avoidance in a significant manner. This study did not include an economic evaluation; therefore, the present work represents the first cost-utility analysis of ABCT.

### 4.3. Strengths and Limitations of This Study

The main strengths of the present work were the randomized design, the delivery of face-to-face relaxation as active control arm, and the low attrition rates. However, the following shortcomings in this economic evaluation should be acknowledged: First, we were faced with a small sample size which limits the generalizability of the findings. This also meant that there were limitations in statistical power for the purposes of detecting significant differences in specific costs between the two treatment arms. Second, the effects and healthcare costs were measured in the short time span of three months. Therefore, it is difficult to draw reliable conclusions about the cost-utility of ABCT when a longer period is considered. Third, the amount of time required for practicing the compassion techniques at home might discourage FM patients and affect long-term adherence. Longitudinal studies with longer follow-up periods would address this issue. Fourth, we used self-report measures for assessing quality of life (EQ-5D-3L) and medical costs (CSRI). Despite empirical evidence indicating that service use data obtained by self-report has equal validity to register collected data [42], it is well-known that self-report measures are subject to recall bias and are obviously less accurate compared to data collected directly from public registers. Fifth, access to ABCT is currently very limited, mostly due to lack of competent therapists with sufficient experience. Sixth, direct costs were probably underestimated in our trial because of the absence of direct non-healthcare costs (e.g., patients’ travel costs to the general practices or temporary hired caregivers). Future studies should collect these types of costs as well as indirect costs related to loss of efficiency at work (absenteeism and presentism) or household work. Larger studies testing the efficiency of ABCT might overcome the methodological difficulties involved when measuring and analyzing indirect costs, such as the high dispersion of the data. Seventh, the non-inclusion of a usual care or waiting-list arm could invite the criticism that the reduced costs and increased quality of life might be achieved in the absence of a specific treatment. The reason for the absence of a treatment as usual arm is that we followed Öst’s recommendations [43] for research on new cognitive behavioral therapies, who encouraged the use of active treatments as comparators. We admit that relaxation is not considered a well-established treatment for FM, but at least it represents a “bona fide” active treatment comparator with equivalent treatment dose. Finally, in complex healthcare interventions such as ABCT—that is, composed of a heterogeneous package of techniques—longitudinal studies with dismantling research designs are also necessary in order to assess which specific ingredients are critical drivers of their efficacy and efficiency. Thus, a pending issue is to elucidate which ABCT sessions or techniques account for the reported increases in quality of life and reductions in healthcare costs.

## 5. Conclusions

To sum up, taking into account the study limitations—inclusion of a small sample size per study arm, the lack of properly trained therapists offering ABCT (which clearly avoids expansion of this therapy), and the short length of the follow-up period—our results should be interpreted with caution. In order to better evaluate the efficiency of ABCT for a public health system or society, in general, a longer study of at least six months to one year would be needed. Undoubtedly, further robust studies are needed to establish whether ABCT truly has a level of cost-utility that would impact policy. Well-designed and better-powered economic evaluations using RCTs with long-term follow-ups are needed to confirm our promising results. Direct comparisons among different forms of CBIs and with other third-wave and fourth-wave therapies seem necessary to disentangle which of these treatments are not only more effective but also more efficient for the healthcare systems and society as a whole.

## Figures and Tables

**Figure 1 jcm-09-00726-f001:**
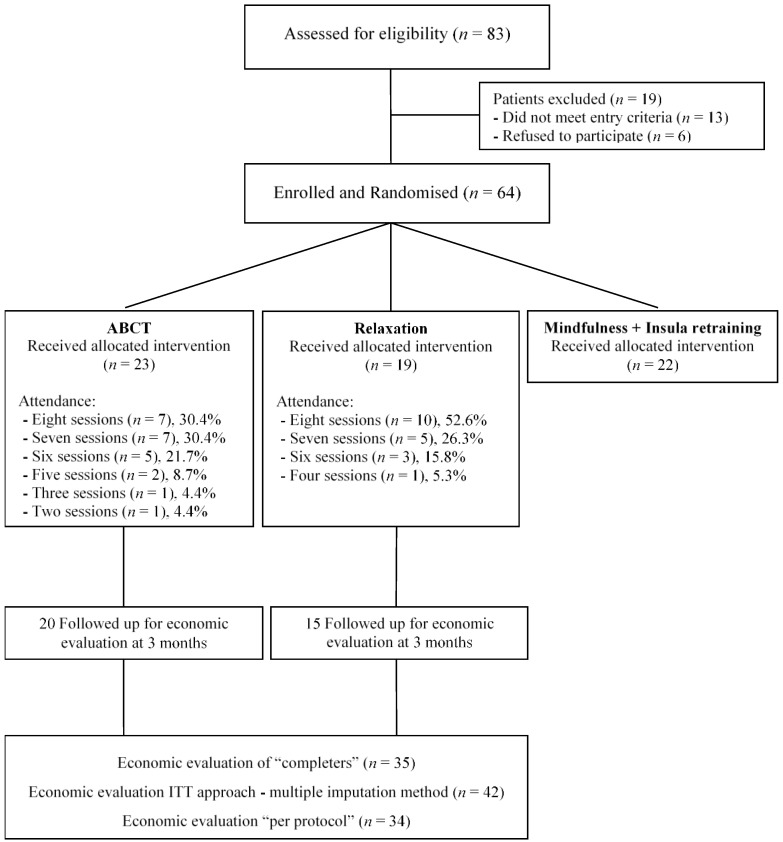
Flowchart of participants in the economic evaluation.

**Table 1 jcm-09-00726-t001:** Unit costs used in the calculations of direct costs (financial year 2016; values in €).

Service (Unit)		Costs (€)
Healthcare	General practitioner (per appointment)	36.97
	Nurse/psychiatric nurse (per appointment)	34.13
	Social worker (per appointment)	35.78
	Clinical psychologist (per appointment)	45.06
	Psychiatrist (per appointment)	45.06
	Other medical specialists (per appointment)	43.82
	Accident and Emergency in hospital (per attendance)	99.34
	Hospital stay (per night)	112
	Diagnostic tests (range)	6.13–455.53
	Pharmacological treatment (per daily dose) *	Various
	Attachment-Based Compassion Therapy (ABCT) and Relaxation (REL) (per participant per group session)	35

Note: Unit costs were applied to each resource use to compute the total cost of resources used by each participant. All unit costs were for the year 2016 in the Autonomous Community of Aragon (Spain). * The cost of prescribed medications was calculated by determining the price per milligram according to the Vademecum International (Red Book; edition 2016) and included the value-added tax.

**Table 2 jcm-09-00726-t002:** Baseline sociodemographic and clinical features of FM patients by study arm [15].

Baseline Characteristics	ABCT (*n* = 23)	REL (*n* = 19)	Test Statistic	*p*
Gender, % female	23 (100)	19 (100)	(*)	1.00
Age (M, SD)	50.83 (8.70)	52.21 (5.95)	T = 0.56 (40)	0.56
Marital status, % with partner	18 (78.3)	13 (68.4)	(*)	0.50
Dwelling, % own home	21 (91.3)	17 (89.5)	(*)	1.00
Educational level				
Primary	10 (43.5)	4 (21.1)	(*)	0.31
Secondary	8 (34.8)	8 (42.1)		
University	5 (21.7)	7 (36.8)		
Work status				
Housework	10 (43.5)	6 (31.6)	(*)	0.54
Employed	4 (17.4)	3 (15.8)		
Sick leave/inability	7 (30.4)	5 (26.3)		
Unemployed	2 (8.7)	5 (26.3)		

Note: ABCT: Attachment-Based Compassion Therapy; REL: Relaxation; *p*: *p*-value associated with the comparison. * Fisher test.

**Table 3 jcm-09-00726-t003:** Mean (SD) costs by cost category and treatment group (complete, unimputed cases).

Baseline (*n* = 42) Time Frame: Last 12 Months	ABCT (*n* = 23) M (SD)	REL (*n* = 19) M (SD)	*p*	Cohen’s d *
Costs (€)				
Primary healthcare services	499.7 (325.1)	441.1 (377.6)	0.82	0.17
Specialized healthcare services	1037.1 (957.4)	897.4 (940.1)	0.79	0.15
Medical tests	367.3 (378.8)	510.8 (636.4)	0.23	0.29
FM-related medications	748.8 (780.4)	520.3 (485.9)	0.62	0.35
Direct costs	2653.0 (1357.1)	2369.6 (1948.8)	0.84	0.18
Outcomes				
EQ-5D Utility score	0.61 (0.17)	0.52 (0.23)	0.43	0.46
**Follow-up (*n* = 35)** **Time frame: last 3 months**	**ABCT (*n* = 20)** **M (SD)**	**REL (*n* = 15)** **M (SD)**	***p***	**Cohen’s d ***
Costs (€)				
Primary healthcare services	83.8 (33.9)	119.0 (40.6)	0.11	0.98
Specialized healthcare services	116.7 (65.4)	247.3 (120.9)	0.57	1.45
Medical tests	36.8 (64.2)	108.8 (114.1)	0.81	0.83
FM-related medications	108.3 (117.8)	151.3 (125.6)	0.75	0.35
Interventions (ABCT – REL)	313.6 (41.1)	305.2 (30.4)	0.00	0.23
Direct costs	659.2 (164.4)	931.6 (217.5)	0.30	1.49
Outcomes				
EQ-5D Utility score	0.72 (0.14)	0.51 (0.25)	0.30	1.11
QALY (based on EQ-5D utility score)	0.17 (0.04)	0.13 (0.06)	0.49	0.83

* Effect size for each pairwise comparison. The rule of thumb is 0.20 = small, 0.50 = medium, and 0.80 = large.

**Table 4 jcm-09-00726-t004:** Incremental cost, effect, and cost-effectiveness ratios from the healthcare perspective.

	Incremental Cost Mean (95% Bootstrap CI)	Incremental Effect Mean (95% Bootstrap CI)	ICER ABCT vs. REL
Completers (*n* = 35) EQ-5D (QALY)	−194.1 (−450.3, 356.1)	**0.023** (0.010, 0.141)	ABCT dominant
ITT (*n* = 42) EQ-5D (QALY)	−256.3 (−447.4, −65.3)	**0.021** (0.009, 0.033)	ABCT dominant
Per protocol (*n* = 34) EQ-5D (QALY)	−168.3 (−499.7, 548.8)	0.003 (−0.013, 0.071)	ABCT dominant

Note: Significant values (*p* < 0.05) in bold. ICERs are cost (in €)/QALY points gained.

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
