# Peer review of "Cost-Utility of Attachment-Based Compassion Therapy (ABCT) for Fibromyalgia Compared to Relaxation: A Pilot Randomized Controlled Trial"

_jcm, 2020, doi:10.3390/jcm9030726_

Round 1

Reviewer 1 Report

Evidence on the efficacy of ABCT was weak. The authors cited a study that followed the same study design and that seemed to have the same sample size. Are there other studies showing the efficacy of ABCT? Due to the nature of the study, the authors had stochastic data available. However, they did not allow for any uncertainty and there was no sensitivity analysis. The authors could have used traditional one-way sensitivity analyses. The authors need to report total intervention costs. The authors should consider including a secondary viewpoint that includes other relevant costs, which may affect results. 

Please further emphasize caution when interpreting your results given your small sample size and length of evaluation. This needs to be addressed specifically in the conclusion.  The authors could provide more findings between the two groups so other findings not related to the economic evaluation are available. The paper it's interesting, but the economic evaluation and the reporting of findings need to be enhanced. Would it be possible to use do nothing as the comparator instead?

In this RCT, the authors investigated the cost-utility of attachment-based compassion (ABCT) therapy and relaxation as adjunctive therapies along with appropriate treatment for patients with fibromyalgia. They used the viewpoint of the health-care system. Competing alternatives are described referencing other papers that provide further detail. A do-nothing (or treatment as usual only) alternative was not included. Evidence on the efficacy of ABCT was weak. The authors cited a study that followed the same study design and that seemed to have the same sample size.

Health care costs were deemed attributable to fibromyalgia and the delivery of the two adjunct therapies. Costs need to be reported clearer, with explicit final total costs. The effect size at baseline should also be reported.

Because this intervention was conducted in a short period of time (less than 12 months) no discounting was needed. However, it would be interesting to see the costs and effects beyond 1 year. Ideally the time horizon in an economic evaluation it’s long enough to capture all significant costs and benefits, including long term adverse effects and repeat therapy. Although an RCT may not be feasible to extend over many years, an ideal economic evaluation would project costs and benefits into the future.

Due to the nature of the study, the authors had stochastic data available. However, they did not allow for any uncertainty and there was no sensitivity analysis. The authors could have used traditional one-way sensitivity analyses. I am worried their results are going to be generalised and misinterpreted. I don’t think the authors findings could be used to inform policymaking, as they claim, based on the study design. The authors could provide more findings between the two groups so that a doctor with no interest in economic evaluation could also see key findings. The paper it's interesting, but the economic evaluation was conducted in a poor manner.

The authors placed their results in context by comparing with other studies. However, they seem to not be mindful of making a comparison between substantially different studies that follow different study designs (different study viewpoint or perspective, different disorders) which may impact the effectiveness of the intervention.  The authors briefly discussed the lack of generalisability of their findings. The authors concluded that ABCT was cost-effective when compared to usual relaxation practices providing value for money. However, they did not discuss implementation issues and limitations due to the study design.

The authors need to add a sensitivity analysis, better report pre and post-intervention effects, total intervention costs, EQ-5D findings and clarify their discussion and conclusion.

Author Response

  • First, we would like to express our thanks to reviewer 1 for his/her exhaustive and insightful review of the manuscript. We addressed the most important points as follows:
  • We admit that research on the ABCT protocol is still in its infancy. Notwithstanding, new evidence in favor of ABCT in a non-clinical sample has been reported. Navarro-Gil and colleagues (2020) carried out a non-randomised RCT with an intervention group (ABCT) and a waiting list (WL) control group. ABCT was significantly more effective than WL for increasing self-compassion, scores on mindfulness facets, and secure attachment style. In addition, ABCT managed to reduce psychological distress and experiential avoidance in a significant manner. We comment this study on page 10.
  • We did not include passive study arms in the RCT. There is an extensive use (and abuse) of treatment as usual or waiting-list as comparator groups in trials of psychological treatments. Following Öst’s (2014) recommendations for future research on third-wave cognitive behavioral therapies, we did use an active treatment as comparator. We admit that relaxation is not considered a well-established treatment for the syndrome in question, but at least it represents a “bona fide” active treatment comparator with equivalent treatment dose. We comment this issue on page 11.
  • Following your suggestion, we have highlighted with more emphasis the preliminary nature of the trial in the conclusions section (see page 11):

To sum up, taking in to account the aforementioned main limitations - inclusion of a small sample size per study arm, the lack of properly trained therapists offering ABCT (which clearly avoids expansion of this therapy), and the short length of the follow-up period -, our results should be interpreted with caution. In order to better evaluate the efficiency of ABCT for a public health system or society in general a longer study of at least six months to one year would be needed. Undoubtedly, further robust studies are needed to establish whether ABCT truly has a level of cost utility that would impact policy. Well-designed and better-powered economic evaluations using RCTs with long-term follow-ups are needed to confirm our promising results. Direct comparisons among different forms of CBIs and with other third-wave and fourth-wave therapies seem necessary to disentangle which of these treatments are not only more effective but also more efficient for the healthcare systems and society as whole.

  • We report the effect sizes (Cohen’s d) at baseline and follow-up in the revised version of the manuscript (see Table 3)
  • One sensitivity analysis was conducted because we computed data analyses following two approaches: completers and ITT. Notwithstanding, taking your comment into account, we have decided to compute a “per protocol” sensitivity analysis, that is, analyses were re-computed only with those patients with FM that had attended at least 6 treatment sessions out of 8, with a final sample size of 34 patients. The results were in the same direction. See Table 4.
  • Regarding indirect costs, we have briefly commented their absence on page 11:

Future studies should collect these type of costs as well indirect costs related to loss of efficiency at work (absenteeism and presentism) or household work. Larger studies testing the efficiency of ABCT might overcome the methodological difficulties involved when measuring and analyzing indirect costs, such as the high dispersion of the data.

Reviewer 2 Report

The manuscript presents the results of a comparative cost-utility analysis of two psychological interventions used in the treatment of fibromyalgia. I find the paper outstanding in its structure and in the use of best practices in clinical research reporting. I wish there were more papers like this taking the recommendations in the field to heart.

I also think that the topic of the paper is relevant for the readership of the journal and that the paper presents a unique contribution to the field and this is highly citable.

The only thing that is really lacking in the paper in its present form is transparency about the research plan, the analysis code, and the research data itself. See my recommendations below:

  • I am not sure what this sentence means: “However, 40 the pilot nature of this RCT limits the generalizability of the findings.” “pilotness” is not a limitation. The limitation is either small sample size or exploratory analysis. This sentence could be clarified or replaced.
  • The Cost-utility analysis described in this paper was not pre-registered in the ClinicalTrials.gov registry, so this should be considered an exploratory analysis. This might be mentioned in the 2.1. Design section. The imputation strategy used in the ITT cases are especially vulnerable to “researcher flexibility”. Thus, without preregistration, results of these analysis should be treated cautiously. This could be mentioned in the limitations section.
  • Transparency could be further improved if the data used in the analysis would be shared together with the analysis code/syntax of the analyses conducted in SPSS and STATA. This would greatly increase the analytical reproducibility of the research study. If the raw data is not sharable as a supplement for some reason, it would be still important to include a data availability statement in the paper that clearly states whether data can be obtained, the exact process for obtaining the data, any criteria or restrictions on who can access the data, and conditions for data reuse. I would suggest that the data should be deposited in a trusted third party repository such as Open Science Framework or GitHub to preserve the longevity of the data.

Author Response

  • We are very thankful to reviewer 2 for his/her kind comments about the quality of our study.
  • Following your suggestions, the sentence “However, the pilot nature of this RCT limits the generalizability of the findings.” has been replaced by this one: “However, the small sample size and short follow-up period limits the generalizability of the findings”.

The main limitations of our pilot trial have been more clearly detailed in the revised version of the manuscript. See page 11.

We admit the absence of a description of the C/E analysis in the registration record. It is a shortcoming that we have mentioned in section 2.1. - Specifically, the cost-utility analysis reported here was not registered in ClinicalTrials.gov, therefore this should be considered an exploratory analysis. For transparency and analytical reproducibility purposes, SPSS data and STATA syntax can be accessed at OSF: https://osf.io/zfjcr/ -

Round 2

Reviewer 1 Report

Dear Authors, 

You have done a great job addressing my concerns.

Before your manuscript it's ready to be published, please remove the word aforementioned from your conclusion (line 386), as it is redundant to use "aforementioned" and then mention the limitations again. 

"To sum up, taking in to account the study limitations - the inclusion of a small
sample size per study arm, the lack of properly trained therapists offering ABCT (which clearly avoids expansion of this therapy), and the short length of the follow-up period -, our results should be interpreted with caution "

Author Response

Fixed. We have removed the word "aforementioned" from the conclusions (see p. 11; line 386)